# Chemerin and Chemokine-like Receptor 1 Expression Are Associated with Hepatocellular Carcinoma Progression in European Patients

**DOI:** 10.3390/biomedicines11030737

**Published:** 2023-03-01

**Authors:** Florian Weber, Kirsten Utpatel, Katja Evert, Oliver Treeck, Christa Buechler

**Affiliations:** 1Institute of Pathology, University of Regensburg, 93053 Regensburg, Germany; 2Department of Gynaecology and Obstetrics, University Medical Center Regensburg, 93053 Regensburg, Germany; 3Department of Internal Medicine I, Regensburg University Hospital, 93053 Regensburg, Germany

**Keywords:** histologic grade, steatosis, fibrosis, UICC score, immunohistochemistry

## Abstract

The chemoattractant protein chemerin is protective in experimental hepatocellular carcinoma (HCC), and high expression in HCC tissues of Asian patients was related to a favorable prognosis. Studies from Asia found reduced expression of chemerin in HCC compared to para-tumor tissues while our previous analysis observed the opposite. Aim of this study was to correlate chemerin expression in HCC tissues with disease severity of European patients Hepatocyte chemerin protein expression was assessed by immunohistochemistry in HCC tissue of 383 patients, and was low in 24%, moderate in 49% and high in 27%. High chemerin protein in the HCC tissues was related to the T stage, vessel invasion, histologic grade, Union for International Cancer Control (UICC) stage and tumor size. Chemokine-like receptor 1 (CMKLR1) is a functional chemerin receptor. CMKLR1 protein in hepatocytes was low expressed in HCC tissues of 36%, moderate in tissues of 32% and high in 32% of the HCCs. Tumor CMKLR1 was associated with the T stage, vessel invasion, histologic grade and UICC stage. Notably, sex-specific analysis revealed that associations of chemerin and CMKLR1 expression with HCC progression were significant in males but not in females. The tumor chemerin and CMKLR1 protein expression were not related to steatosis, inflammation and fibrosis grades. In summary, chemerin as well as CMKLR1 protein were related to disease severity of European HCC patients, and this was significant in males. This observation is in contrast to Asian patients where higher chemerin in the tumors was protective. Current analysis provides evidence for ethnicity and sex-related differences of tumor expressed chemerin and HCC severity.

## 1. Introduction

Chemerin is a chemoattractant protein and is mainly produced by adipocytes and hepatocytes [1]. Serum chemerin protein is increased in obesity in accordance with its expression in fat tissues. Chemerin was initially identified to function as a chemoattractant for cells expressing chemokine-like receptor 1 (CMKLR1). Later on, a function of chemerin was demonstrated for the insulin response, adipogenesis, and blood pressure control [1,2,3]. There is increasing evidence that locally produced chemerin also has a role in different cancers. Most of the studies having reported an anti-tumorigenic function of chemerin also identified a downregulation of the chemerin protein in the tumors [4,5]. 

The first study showing anti-tumor properties of chemerin was in melanoma. Here, the chemerin-induced recruitment of natural killer cells was protective [6]. In experimental hepatocellular carcinoma (HCC), chemerin reduced tumor growth and metastasis by affecting the T-cell function. Chemerin did not change the proliferation of hepatocyte cell lines [7,8,9]. Chemerin, moreover, increased the expression and phosphatase activity of phosphatase and tensin homolog (PTEN) and thereby suppressed HCC metastasis [8].

Accordingly, chemerin protein was found to be low in HCC tissues. A study from China published in 2011 showed that chemerin protein analyzed by immunohistochemistry was reduced in HCC tissues for about 60% of the patients [10]. Chemerin tumor protein positively correlated with immune cell infiltration. Patients with high intratumoral chemerin had a better prognosis [10]. The association of higher chemerin tumor with less disease severity and more favorable prognosis was confirmed by a second study that was also conducted in China. Notably, chemerin protein was also detected by immunoblot and was again reduced in the HCC tissues [9]. This was also described by a third study from China [8]. 

In contradiction with these findings from China, our previous analysis of chemerin protein levels by immunoblot in HCC tissues obtained from European patients detected higher proteins in HCC than the para-tumor tissues [8,9,10,11]. HCC-related upregulation of chemerin was influenced by disease etiology, and was observed in non-alcoholic fatty liver disease (NAFLD) and hepatitis B virus (HBV) infection, but not in hepatitis C virus (HCV)-positive patients. Tumor-localized chemerin protein was not related to the T stage in this small cohort of European patients [11]. The upregulation of chemerin in the HCC tissues of Europeans suggests a tumor-promoting function, but associations of hepatocyte-expressed chemerin and HCC progression in this population have not been studied so far. 

CMKLR1 is expressed by almost all the cell types analyzed [4,12,13,14,15]. G protein-coupled receptor 1 (GPR1) is the second chemerin receptor, but it was less well-studied [13,16]. CMKLR1 protein is expressed by primary human hepatocytes and various hepatocellular cell lines [8,17]. CMKLR1 protein was found to be reduced in HCC tissues in comparison to the tumor adjacent tissues of European patients with NAFLD. In patients with viral hepatitis, the tumor and para-tumor expression of CMKLR1 protein was similar [11]. A lower expression of CMKLR1 in the tumor tissues of NAFLD patients may prevent chemerin from exerting its anti-tumor effects. 

It has to be noted that chemerin is released by the cells as inactive prochemerin and cleavage of the carboxyl terminus by serine proteases results in activation [1]. An analysis of chemerin protein levels by commercial antibodies cannot discriminate the different chemerin isoforms, and it is unknown whether chemerin protein in HCC tissues is biologically active. HCC protective effects of chemerin are, however, attributed to the biologic active chemerin isoforms [1,4] 

Total chemerin protein levels in the tumors are, however, of diagnostic and prognostic value in HCC, at least in patients from China [8,9,10]. The current analysis sought to examine the relationships between hepatocyte-expressed chemerin in HCC and disease severity in a large cohort of patients from Europe. CMKLR1 is a functional chemerin receptor, and its protein expression in hepatocytes was analyzed in parallel. 

## 2. Materials and Methods

### 2.1. Patients

HCC tissues of 383 patients (315 males and 68 females) were obtained from the years 2000 to 2021. The patients are from the eastern part of Bavaria and about 2.5% of the inhabitants of Germany are from Asian countries [18]. Mean age of the patients was 64.32 ± 11.48 years. 

Seven tissue microarrays (TMAs), in which up to 60 separate tissue cores per TMA were assembled, were prepared using standard techniques already described [19]. Experienced pathologists evaluated hematoxylin and eosin stained sections of HCC tissues and selected representative areas. One core was used from each of the tumors and included in the final TMA with about 60 specimens in each paraffin block. The TNM classification system was used to define pathological primary tumor extent (pT stage) and disease stage according to the Union for International Cancer Control (UICC) staging system. WHO guidelines were applied for histological tumor grading [20]. 

This was a retrospective study, which was conducted in accordance with the Declaration of Helsinki. Approval for this study was obtained by the Ethics Committee of the University Hospital of Regensburg. The Ethics Committee confirmed that an informed consent was not needed for this retrospective study (protocol code 22-2788-101; date of approval: 23 February 2022). 

### 2.2. Immunohistochemistry

The IHC-plus^TM^ RARRES2/chemerin antibody (order number: LS-B13333) and the IHC-plus^TM^ CHEMR23/CMKLR1 antibody (order number: LS-B12924) were obtained from Biozol (Eching, Germany) and diluted 1:100 fold for analysis. The IHC-plus™ antibodies are tested in immunohistochemistry against human formalin-fixed paraffin-embedded tissues and have an excellent specificity and sensitivity for detecting the target protein [21]. The staining protocol was established as is conducted for all antibodies in the Institute for Pathology by trying out different dilutions of the antibody and different pre-treatments on a control TMA with the most important normal human tissues until the optimal staining protocol with a balance between staining intensity of the desired protein and minimal background staining is found.

For immunohistochemistry, 4 μm thick sections of the TMA blocks were deparaffinized, treated at 120 °C for 5 min, and incubated with Tris-EDTA buffer (pH 9). Endogenous peroxidase was blocked with peroxidase blocking solution (Dako, Glostrup, Denmark), and antibody incubation was performed. All antibodies were incubated for 30 min at room temperature. Staining was performed using the Dako EnVision™^+^ Detection System, Peroxidase/DAB+, Rabbit/Mouse (Dako, Glostrup, Denmark). After these incubations, the slides were counterstained with hematoxylin.

Immunohistochemical staining was independently assessed by two expert pathologists (FW und KE); divergent results were discussed and a consensual score was reached. Both antibodies did not stain the tissues when the primary antibodies were not added [22]. Both antibodies showed unspecific nuclear staining of lymphocytes [22], and therefore, only cytoplasmic and membranous staining of hepatocytes was considered for the scoring. Nuclear staining was not included in the assessment. 

Low staining intensity was defined as no or barely visible membranous and/or cytoplasmic staining; for high staining intensity, either heterogeneous or patchy membranous and/or cytoplasmic staining comparable to that of bile ducts in normal liver tissue was considered, and medium staining intensity was defined as any staining intensity between the other two groups. 

A three-tiered scoring system was used for chemerin, with a score of 1 denoting low cytoplasmic and/or membranous staining, 2 denoting moderate staining, and 3 denoting high staining. A three-tiered score system was also used to define the level of CMKLR1 protein expression, with scores 1, 2, and 3 designating low, moderate, and high cytoplasmic and/or membranous staining, respectively.

Immunohistochemical data traditionally are semi-quantified by pathologist visual scoring of the staining intensity [23]. Accordingly, Li et al. scored the staining intensity of chemerin in HCCs on a scale of 0–3 (0: no staining, 1: weak intensity, 2: moderate intensity, and 3: strongest intensity) [8] and Lin et al. scored the chemerin staining as 0 to 3+ [10]. A separate study counted the staining-positive cells and measured chemerin density by Image-Pro Plus v6.2 software [9]. Whether nuclear staining has not been observed or was not quantified is not discussed in these manuscripts [8,9,10]. Here, cytoplasmic and membrane staining but not nuclear staining of hepatocytes was evaluated, which is not possible with automated measurements. 

The control group consisted of non-neoplastic liver tissue from patients suffering from HCC.

### 2.3. Histological Scores

Histological assessment of tumoral steatosis, grade of inflammation, and liver fibrosis was performed by expert liver pathologists (K.U. and K.E.). For grade of tumoral steatosis, the percentage of tumoral fat vacuoles in relation to total tumor volume was stated. Steatosis grade ranged from 0% to 80%. 

Intratumoral inflammation was graded on a 4-tiered scale from 0–3, where 0 refers to no inflammation and 3 describes severe inflammation. Fibrosis of non-tumoral liver parenchyma in the surgical specimen was graded according the Ishak fibrosis score on a 7-tiered scale from 0–6, where 0 stands for no fibrosis and 6 is equal to complete cirrhosis [24]. 

### 2.4. Statistics

Data are given as mean value ± standard deviation (SD). Statistical tests used were Mann–Whitney U Test (to test for differences between two independent groups), and Kruskal–Wallis Test (to test for differences between three independent groups) (SPSS Statistics 26.0 program). The *p*-values were corrected for multiple comparisons by Bonferroni. A *p*-value < 0.05 was considered significant. 

## 3. Results

### 3.1. Chemerin Expression in HCC Tissues

Chemerin protein was analyzed by immunohistochemistry and was found to be expressed low in 24%, moderate in 49%, and high in 27% of the 383 HCC tissues analyzed (Figure 1). 

Stratification of the patients into low, moderate, and high chemerin-expressing tumors showed that the T stage was significantly higher in the latter group in comparison to the low and moderate chemerin group. Blood vessel invasion was increased in HCCs with high compared to low chemerin expression. Histologic grading and tumor size were significantly higher in patients with moderate or high chemerin tumor protein compared to those with low expression. The Union for International Cancer Control (UICC) stage was higher in patients with moderate and high chemerin expression compared to patients with low expression levels. Lymph node invasion and age did not differ between the three groups (Table 1).

Notably, chemerin protein was not related to steatosis score, inflammation, or fibrosis grade (Table 1). CMKLR1 expression increased in parallel with chemerin (Table 1).

### 3.2. Sex-Specific Differences of Chemerin Expression in HCC Tissues

The chemerin staining scores of the tumors of females and males did not differ (*p* = 0.156). The T stage (*p* = 0.961), lymph node invasion (*p* = 0.919), and vessel invasion (*p* = 0.975) were similar between sexes. Tumors of females were 8.51 ± 6.09 cm and of males 5.95 ± 4.41 cm, and so, they were larger in females (*p* = 0.001). A sex-specific analysis showed that the chemerin protein in the HCC tissues of females was not related to the T stage, lymph node invasion, vessel invasion, grading, tumor size, or UICC score (Table 2).

Chemerin protein in the HCC tissues of the female patients was not related to the steatosis score, inflammation, or fibrosis grade (Table 2). The age of the patients did not differ between HCCs with low, moderate, and high chemerin (Table 2). As was shown for the whole cohort, CMKLR1 protein increased in parallel with chemerin (Table 2). 

In the male patients the T stage was significantly higher in the high chemerin group in comparison to the low and moderate chemerin group. Blood vessel invasion and tumor size were increased in HCCs with a high compared to low chemerin expression. Histologic grading was significantly higher in patients with moderate or high chemerin tumor protein compared to HCCs with low expression. The UICC score was significantly increased in the patients with high chemerin compared to those with moderate and low expression in the tumors. Lymph node invasion did not differ between the three groups (Table 3).

Notably, chemerin protein in the HCC tissues of males was not related to age, steatosis score, inflammation, or fibrosis grade (Table 3). CMKLR1 protein scores increased in parallel with the chemerin scores (Table 3). 

### 3.3. CMKLR1 Expression in HCC Tissues

Notably, hepatocyte-expressed CMKLR1 was mostly found in the cytoplasm. The CMKLR1 protein of 382 patients could be judged and CMKLR1 was lowly expressed in 36%, moderately expressed in 32% and highly expressed in 32% of the HCCs (Figure 2). 

Comparison of the patients with low, moderate, and high tumor CMKLR1 proteins revealed that the T stage and blood vessel invasion were significantly increased in the high versus the low CMKLR1 group. Histologic grading was significantly increased in the high CMKLR1 group in comparison to the low and to the moderate group. The UICC stage was highest in the patients with high CMKLR1 in comparison to the two other groups (Table 4). 

CMKLR1 protein was not related to tumor size, steatosis score, inflammation, or fibrosis grade. Age was not a confounding factor (Table 4). 

Chemerin protein expression increased with higher CMKLR1 protein (Table 4). 

### 3.4. Sex-Specific Differences of CMKLR1 Expression in HCC Tissues

The CMKLR1 protein expression scores did not differ between males and females (*p* = 0.077). Sex-specific analysis showed that the CMKLR1 protein in the HCC tissues of females was not related to the T stage, lymph node invasion, vessel invasion, grading, tumor size, or UICC score (Table 5).

The CMKLR1 protein in the HCC tissues of the female patients was not related to age, steatosis score, inflammation, or fibrosis grade (Table 5). 

The chemerin protein was higher in the high CMKLR1 compared to the low CMKLR1 group (Table 5). 

Comparison of the male patients with low, moderate, and high tumor CMKLR1 protein revealed that the T stage, lymph node invasion, and vessel invasion were significantly increased in the high versus the low CMKLR1 group. Histologic grading and the UICC stage were significantly increased in the high CMKLR1 group in comparison to the low and to the moderate group (Table 6). 

The CMKLR1 protein was not related to tumor size, steatosis grade, fibrosis grade, or age (Table 6). Patients with high CMKLR1 had more inflammation than those with moderate CMKLR1 (Table 6). 

Chemerin protein expression was elevated in the moderate and high CMKLR1 groups in comparison to the low CMKLR1 group (Table 6). 

Disease etiology was documented for 63 patients, and 19 patients had HBV and 44 patients had HCV. However, chemerin (*p* = 0.188) and CMKLR1 protein in the tumors (*p* = 0.600) did not differ between the two groups. 

## 4. Discussion

The current study showed that chemerin and CMKLR1 protein expression in the tumors of European HCC patients are associated with disease progression. This is in contrast to Chinese patients, where the tumor chemerin protein levels were found to decline with disease severity [8,9,10]. Sex-specific analysis, which has not been performed in the Asian studies, suggests that strong associations of chemerin and CMKLR1 protein with HCC stages exist in males but not females. As far as we are aware, there have never been reports of opposing associations between the expression of a protein and the severity of HCC in Asian and European patients. 

The three studies from Asia consistently described that chemerin is reduced in HCC tissues in comparison to the peritumoral tissue [8,9,10]. In contrast, it was shown in our previous analysis that chemerin protein expression is increased in the HCC tissues of European patients. In comparison to tumor-adjacent tissues, chemerin protein was found to be induced in the tumors of European HCC patients with NAFLD and HBV, but not in HCV-related HCC [11]. Associations of the tumor chemerin expression with the T stages were not identified in these relatively small cohorts [11].

In Asian patients, significant negative correlations of chemerin with tumor size and histological grade have been detected [10]. A second analysis could, however, not observe the associations of chemerin protein with the tumor size, tumor character, such as single nodule or multiple nodule, or TNM stage [8]. In our European cohort, chemerin protein in the HCC tissues was associated with tumor stage, grading, tumor size, and vessel invasion, and thus, UICC staging. These relations could be verified in the male patients. Whether equivalent associations exist in females requires further study. In the 68 females of our cohort, no such relations were observed. There are much fewer females than males in our cohort in accordance with the two to four fold higher HCC incidence of males [25] and the study group may be too small to show significance. In contrast to what has been published [25], females had larger tumors than males, and this is specific to our study group. 

In Asian patients, macrophage numbers were higher in HCC tissues with increased chemerin expression [8]. The dendritic cell and natural killer cell counts were positively related to chemerin protein [10]. Associations of tumoral inflammation and chemerin protein expression were not identified in the current cohort. Chemerin is a chemoattractant for macrophages, dendritic cells, and natural killer cells. To function as an immune cell attractant, chemerin has to be C-terminally processed [1]. Positive correlations of chemerin protein with macrophage, dendritic cell, and natural killer cell numbers suggest that these active isoforms are available in the HCC tissues of Chinese patients. In the HCC tissues of European patients, chemerin protein did not correlate with the inflammation score and thus seems to be inactive. Commercial antibodies cannot discriminate between the different chemerin isoforms, and which of the chemerin variants are abundant in HCC tissues has not been evaluated so far. 

Currently, there is no explanation for the discordant findings between European and Asian patients. Genetic mutations and signatures vary between patients with different disease etiologies, and ethnic factors also play a role here [26]. Thus, Sal-like protein 4 (SALL4) was found re-expressed in tissues of about 50% of Chinese HCC patients, and serum levels were increased. In these patients, serum SALL4 levels were related to tumor recurrence and survival [27]. SALL4 upregulation was, however, very rare in Western HCC patients [28]. 

The underlying liver disease etiologies of HCC differ between Western and Asian countries [26]. Chronic viral infections and NAFLD are risk factors for HCC, and the epidemiology of HCC varies with geographic location. In Asia, nearly 80% of liver cancers are caused by chronic HBV infection, 3% by HCV, and 1% by NAFLD [29]. In Germany, 20% of HCCs are related to chronic HCV, 25% to chronic HBV, and 22% to NAFLD [30]. 

A detailed comparison of hepatic chemerin expression between patients with different causes for chronic liver injury has not been performed so far. Interestingly, HBV infection was found to lower the chemerin protein in HCC tissues of Asian patients and had the opposite effect in European patients [8,11]. This shows a contrary regulation of chemerin protein in tumor tissues of HBV-infected Chinese and European patients. Differences in disease etiology of Chinese and European HCC patients alone can, therefore, not explain the opposite association of HCC-expressed chemerin protein with disease severity. 

Chemerin protein in the HCC tissues of Chinese patients was not changed with sex and age [10], and associations of the protein levels with sex and age were not identified in the patient samples studied here. Interestingly, associations of chemerin expression in the HCC tissues with disease progression seem to be less pronounced in females, and no significant relations were found in the cohort studied here. 

In the Chinese cohort, the chemerin tumor protein was not related to cirrhosis [10], which is in accordance with the current findings in the European patients. Moreover, tumoral steatosis was not associated with altered chemerin levels in the Western patients. Liver steatosis is caused by HCV infection and alcohol abuse and is commonly observed in obesity [31,32]. Data regarding chemerin expression in the steatotic human liver are not concordant [1], and yet to be identified factors besides disease etiology seem to have a role herein. 

Studies from Europe and Japan have shown that serum chemerin is low in patients with liver cirrhosis and that it correlates with hepatic dysfunction [33,34,35]. Although this suggests reduced hepatic chemerin protein expression in the cirrhotic liver, chemerin protein expression in the HCC tissues was not related to fibrosis scores. Liver dysfunction thus may impair chemerin release from the liver and/or fat tissues into the blood and/or enhance its elimination from the body. 

Considering the close association of serum chemerin with hepatic dysfunction, it is not unexpected that studies measuring circulating chemerin in HCC have revealed discordant results. In the European patients, serum chemerin did not differ between HCC and controls [36]. A study from Japan could not find significant correlations of serum chemerin levels with HCC prognosis [35]. A study from China described a nearly 20-fold lower chemerin concentration in the blood of HCC patients compared with healthy subjects [9]. It is well-known that serum chemerin declines in patients with liver cirrhosis, and HCC occurs most often in patients with cirrhosis [33,34,37]. Future research has to compare serum chemerin levels between patients with and without HCC stratified for liver disease severity.

Until now, studies on the hepatic expression of CMKLR1 have been sparse. CMKLR1 protein was found to be positively associated with the T stage, grading, vessel invasion and, UICC stage. Again, these associations were significant in males but not in females. 

The CMKLR1 protein did not change with age and sex, and moreover, it was not related to tumoral steatosis, inflammation, and liver fibrosis.

In males, high CMKLR1 protein was related to more inflammation. This difference existed between HCCs with moderate and high but not between low and high CMKLR1 expression. The corresponding *p* value was rather high, arguing against a strong association of CMKLR1 and inflammation in the HCC tissues of males. 

In hepatocytes, CMKLR1 was mostly localized in the cytoplasm. Notably, this also applies to PTEN [38], which was shown to interact with CMKLR1 [8]. Membrane–cytoplasmic staining of CMKLR1 was described in mice livers [39]. The CMKLR1 antibody distributed by a different company showed cytoplasmic and nuclear staining in kidney carcinoma and stomach carcinoma [40]. Nuclear staining was also observed in the current experiments but was regarded as unspecific and not quantified. It is possible that CMKLR1 localizes to the nucleus in different tumor cells, but further experiments have to prove this assumption. Nuclear staining of chemerin is considered unspecific, and there are no studies that have shown that chemerin translocates to the nucleus as far as we know. 

The limitations of the current study are that survival was not documented and that the disease etiology of most patients was unknown. Although positive associations of chemerin and CMKLR1 protein levels with markers of HCC severity suggest that higher levels of these proteins are linked to a worse prognosis, this has to be experimentally proven. Further limitations are the descriptive nature of this retrospective study, that only immunohistochemistry was used, and that chemerin as well as CMKLR1 expression was semi-quantified by a pathologist visual scoring of the staining intensity. The ethnicity of the patients with HCC was not recorded. The patients enrolled in the study are from the eastern part of Bavaria. About 2.5% of the German population are of Asian descent, and data of these patients could not be removed [18]. 

The studies on Asian HCC patients published so far suggested a protective role of chemerin in HCC [8,9], but the issue of whether chemerin in human HCC tissues is biologically active has not been evaluated. Future research has to consider chemerin activity and to compare the chemerin isoform composition of HCC tissues obtained from Western and Asian patients. Furthermore, the molecular mechanisms underlying the conflictive results of studies on Asian and European patients regarding the association of chemerin with HCC progression remain to be investigated. 

## 5. Conclusions

High chemerin and CMKLR1 proteins in European patients with HCC are related to adverse clinical parameters such as the T stage, vessel invasion, grading, and UICC stage, indicating a tumor-promoting effect of this receptor and its ligand. In Asian patients, the high chemerin expression in tumors was protective. Preliminary analysis, moreover, indicates that the association of chemerin and CMKLR1 with HCC progression was less evident in female patients. The current findings suggest that ethnicity and sex play a role in HCC pathophysiology and have to be considered in the development of diagnostic and therapeutic approaches. 

## Figures and Tables

**Figure 1 biomedicines-11-00737-f001:**
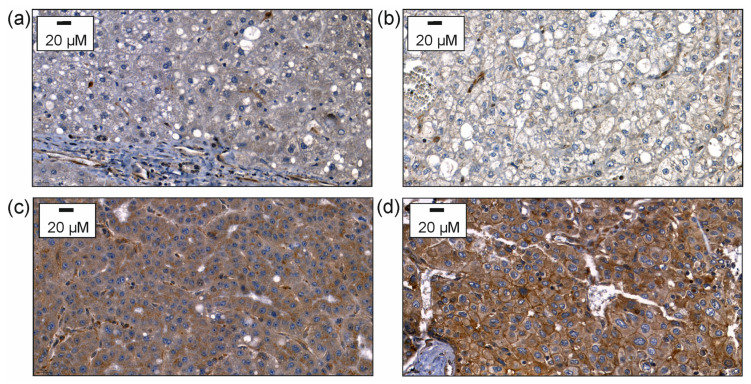
Immunohistochemical expression of chemerin. After staining with the antibody, the slides were counterstained with hematoxylin. (**a**) Immunohistochemical expression of chemerin in non-neoplastic liver tissue of a patient suffering from HCC. (**b**) Immunohistochemical expression of chemerin in HCC tissue with staining of hepatocytes scored as 1 for weak cytoplasmic staining. (**c**) Immunohistochemical expression of chemerin in HCC tissue with staining of hepatocytes scored as 2 for moderate staining; (**d**) Immunohistochemical expression of chemerin in HCC tissue with staining of hepatocytes scored as 3 for strong heterogeneous or patchy cytoplasmic and/or membranous staining.

**Figure 2 biomedicines-11-00737-f002:**
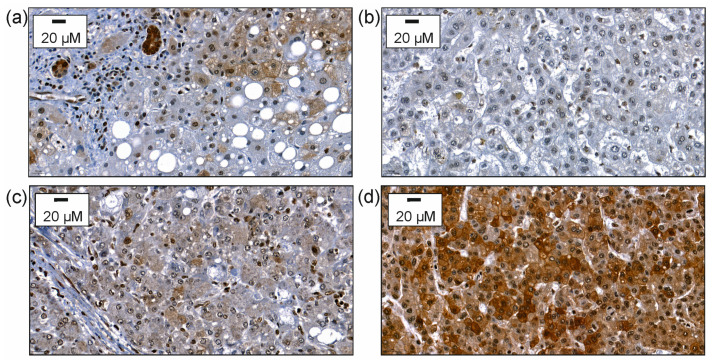
Immunohistochemical expression of CMKLR1. After staining with the antibody the slides were counterstained with hematoxylin. (**a**) Immunohistochemical expression of CMKLR1 in non-neoplastic liver tissue of a patient suffering from HCC. (**b**) Immunohistochemical expression of CMKLR1 in HCC tissue with staining of hepatocytes scored as 1 for weak cytoplasmic staining. (**c**) Immunohistochemical expression of CMKLR1 in HCC tissue with staining in hepatocytes scored as 2 for moderate staining. (**d**) Immunohistochemical expression of CMKLR1 in HCC tissue with staining of hepatocytes scored as 3 for strong heterogeneous or patchy cytoplasmic and/or membranous staining.

**Table 1 biomedicines-11-00737-t001:** T stage, lymph node invasion, vessel invasion, grading, UICC stage, steatosis, inflammation, fibrosis stages, age, and CMKLR1 scores of patients stratified for low, moderate, and high chemerin protein levels in the tumors. a: comparison of tumors with low and high chemerin, b: comparison of tumors with moderate and high chemerin, and c: comparison of tumors with low and moderate chemerin (inv, invasion).

	Low Chemerin	Moderate Chemerin	High Chemerin	*p*-Value
Patients	92	186	105	
T stage	1.69 ± 0.66 ^a^	1.85 ± 0.80 ^b^	2.21 ± 0.88 ^a,b^	<0.001 ^a^, 0.004 ^b^
Lymph node inv	0.03 ± 0.18	0.09 ± 0.29	0.07 ± 0.25	not significant
Vessel inv	0.32 ± 0.53 ^a^	0.50 ± 0.62	0.67 ± 0.69 ^a^	<0.001 ^a^
Grading	1.70 ± 0.62 ^a,c^	1.99 ± 0.72 ^c^	2.10 ± 0.63 ^a^	<0.001 ^a^, 0.003 ^c^
Tumor size (cm)	4.92 ± 3.75 ^a,c^	6.47 ± 4.77 ^c^	7.62 ± 5.50 ^a^	<0.001 ^a^, 0.02 ^c^
UICC score	1.71 ± 0.76 ^a^	1.92 ± 0.86 ^b^	2.22 ± 0.90 ^a,b^	<0.001 ^a^, 0.018 ^b^
Steatosis grade %	14 ± 15	15 ± 19	14 ± 19	not significant
Inflammation grade	0.70 ± 0.57	0.74 ± 0.70	0.85 ± 0.68	not significant
Fibrosis grade	4.30 ± 2.42	3.83 ± 2.41	3.99 ± 2.37	not significant
Age (years)	63.64 ± 10.32	64.7222 ± 12.53	64.19 ± 10.53	not significant
CMKLR1	1.44 ± 0.72 ^a,c^	1.91 ± 0.78 ^b,c^	2.50 ± 0.66 ^a,b^	<0.001 ^a,b,c^

**Table 2 biomedicines-11-00737-t002:** T stage, lymph node invasion, vessel invasion, grading, UICC stage, steatosis, inflammation, fibrosis stages, age, and CMKLR1 scores of female patients stratified for low, moderate, and high chemerin protein levels in the tumors. a: comparison of tumors with low and high chemerin, b: comparison of tumors with moderate and high chemerin, and c: comparison of tumors with low and moderate chemerin (inv, invasion).

	Low Chemerin	Moderate Chemerin	High Chemerin	*p*-Value
Patients	11	36	21	
T stage	1.80 ± 0.79	1.90 ± 0.85	2.17 ± 0.92	not significant
Lymph node inv	0.0 ± 0.0	0.11 ± 0.32	0.0 ± 0.0	not significant
Vessel inv	0.27 ± 0.65	0.56 ± 0.70	0.70 ± 0.80	not significant
Grading	1.64 ± 0.67	2.17 ± 0.70	2.10 ± 0.62	not significant
Tumor size (cm)	7.08 ± 6.98	9.08 ± 6.30	8.28 ± 5.36	not significant
UICC score	1.73 ± 0.79	1.94 ± 0.83	2.05 ± 0.86	not significant
Steatosis grade %	15 ± 15	10 ± 10	10 ± 9	not significant
Inflammation grade	0.80 ± 0.63	0.81 ± 0.82	0.76 ± 0.62	not significant
Fibrosis grade	3.00 ± 3.00	3.00 ± 2.73	3.78 ± 2.44	not significant
Age (years)	61.54 ± 12.88	63.41 ± 13.07	62.57 ± 10.39	not significant
CMKLR1	1.55 ± 0.82 ^a,c^	1.92 ± 0.84 ^b,c^	2.76 ± 0.44 ^a,b^	<0.001 ^a,b,c^

**Table 3 biomedicines-11-00737-t003:** T stage, lymph node invasion, vessel invasion, grading, UICC stage, steatosis, inflammation, fibrosis stages, age, and CMKLR1 scores of male patients stratified for low, moderate, and high chemerin protein levels in the tumors. a: comparison of tumors with low and high chemerin, b: comparison of tumors with moderate and high chemerin, and c: comparison of tumors with low and moderate chemerin (inv, invasion).

	Low Chemerin	Moderate Chemerin	High Chemerin	*p*-Value
Patients	81	150	84	
T stage	1.67 ± 0.65 ^a^	1.84 ± 0.80 ^b^	2.23 ± 0.87 ^a,b^	<0.001 ^a^, 0.006 ^b^
Lymph node inv	0.04 ± 0.19	0.09 ± 0.28	0.09 ± 0.28	not significant
Vessel inv	0.32 ± 0.52 ^a^	0.49 ± 0.60	0. 66 ± 0.67 ^a^	0.002 ^a^
Grading	1.70 ± 0.62 ^a,c^	1.95 ± 0.72 ^c^	2.11 ± 0.64 ^a^	0.001 ^a^, 0.037 ^c^
Tumor size (cm)	4.62 ± 3.00 ^a^	5.83 ± 4.09	7.46 ± 5.55 ^a^	0.002 ^a^
UICC score	1.70 ± 0.77 ^a^	1.92 ± 0.87 ^b^	2.26 ± 0.91 ^a,b^	<0.001 ^a^, 0.012 ^b^
Steatosis grade %	14 ± 16	17 ± 20	15 ± 21	not significant
Inflammation grade	0.68 ± 0.57	0.73 ± 0.67	0.88 ± 0.70	not significant
Fibrosis grade	4.46 ± 2.31	4.01 ± 2.30	4.04 ± 2.37	not significant
Age (years)	63.93 ± 9.98	65.03 ± 12.42	64.60 ± 10.59	not significant
CMKLR1	1.42 ± 0.70 ^a,c^	1.91 ± 0.76 ^b,c^	2.43 ± 0.68 ^a,b^	<0.001 ^a,b,c^

**Table 4 biomedicines-11-00737-t004:** T stage, lymph node invasion, vessel invasion, grading, tumor size, UICC stage, steatosis, inflammation, fibrosis stages, age, and chemerin scores of patients stratified for low, moderate, and high CMKLR1 protein in the tumors. a: comparison of tumors with low and high CMKLR1, b: comparison of tumors with moderate and high CMKLR1, and c: comparison of tumors with low and moderate CMKLR1 (inv, invasion).

Histologic Measures	Low CMKLR1	Moderate CMKLR1	High CMKLR1	*p*-Value
Patients	138	123	121	
T stage	1.76 ± 0.73 ^a^	1.86 ± 0.81	2.12 ± 0.87 ^a^	0.006 ^a^
Lymph node inv	0.04 ± 0.19	0.08 ± 0.28	0.10 ± 0.30	not significant
Vessel inv	0.40 ± 0.59 ^a^	0.50 ± 0.63	0.62 ± 0.67 ^a^	0.015 ^a^
Grading	1.75 ± 0.69 ^a^	1.94 ± 0.65 ^b^	2.20 ± 0.65 ^a,b^	<0.001 ^a^, 0.012 ^b^
Tumor size (cm)	6.02 ± 4.58	6.17 ± 4.77	7.09 ± 5.20	not significant
UICC	1.84 ± 0.81 ^a^	1.88 ± 0.85 ^b^	2.17 ± 0.91 ^a,b^	0.012 ^a^, 0.037 ^b^
Steatosis grade %	19 ± 21	13 ± 16	11 ± 13	not significant
Inflammation grade	0.78 ± 0.61	0.65 ± 0.63	0.87 ± 0.74	not significant
Fibrosis grade	3.90 ± 2.50	4.00 ± 2.48	4.05 ± 2.24	not significant
Age (years)	64.66 ± 12.08	63.88 ± 11.00	64.31 ± 11.37	not significant
Chemerin	1.60 ± 0.61 ^a,c^	2.15 ± 0.62 ^b,c^	2.41 ± 0.67 ^a,b^	<0.001 ^a,b^, 0.013 ^c^

**Table 5 biomedicines-11-00737-t005:** T stage, lymph node invasion, vessel invasion, grading, tumor size, UICC stage, steatosis, inflammation, fibrosis stages, age, and chemerin scores of female patients stratified with low, moderate, and high CMKLR1 protein in the tumors. a: comparison of tumors with low and high chemerin (inv, invasion).

Histologic Measures	Low CMKLR1	Moderate CMKLR1	High CMKLR1	*p*-Value
Patients	21	18	29	
T stage	2.11 ± 0.90	1.65 ± 0.70	2.09 ± 0.90	not significant
Lymph node inv	0.10 ± 0.30	0.06 ± 0.24	0.04 ± 0.19	not significant
Vessel inv	0.48 ± 0.68	0.44 ± 0.70	0.68 ± 0.77	not significant
Grading	1.76 ± 0.77	2.22 ± 0.65	2.17 ± 0.60	not significant
Tumor size	9.19 ± 6.20	8.14 ± 6.88	8.24 ± 5.67	not significant
UICC	2.10 ± 0.83	1.72 ± 0.75	1.97 ± 0.87	not significant
Steatosis grade %	15 ± 13	13 ± 12	9 ± 8	not significant
Inflammation grade	0.78 ± 0.65	0.83 ± 0.79	0.78 ± 0.75	not significant
Fibrosis grade	2.88 ± 2.64	2.38 ± 2.75	4.04 ± 2.49	not significant
Age (years)	65.48 ± 14.73	62.72 ± 8.09	61.03 ± 12.16	not significant
Chemerin	1.66 ± 0.48 ^a^	2.17 ± 0.62	2.48 ± 0.63 ^a^	<0.001 ^a^

**Table 6 biomedicines-11-00737-t006:** T stage, lymph node invasion, vessel invasion, grading, tumor size, UICC stage, steatosis, inflammation, fibrosis stages, age, and chemerin scores of male patients stratified with low, moderate, and high CMKLR1 protein in the tumors. a: comparison of tumors with low and high CMKLR1, b: comparison of tumors with moderate and high CMKLR1, and c: comparison of tumors with low and moderate CMKLR1 (inv, invasion).

Histologic Measures	Low CMKLR1	Moderate CMKLR1	High CMKLR1	*p*-Value
Patients	117	105	92	
T stage	1.70 ± 0.68 ^a^	1.91 ± 0.83	2.13 ± 0.87 ^a^	0.002 ^a^
Lymph node inv	0.03 ± 0.16 ^a^	0.09 ± 0.28	0.12 ± 0.33 ^a^	0.028 ^a^
Vessel inv	0.39 ± 0.57 ^a^	0.51 ± 0.62	0.60 ± 0.63 ^a^	0.028 ^a^
Grading	1.74 ± 0.68 ^a^	1.89 ± 0.64 ^b^	2.21 ± 0.67 ^a,b^	<0.001 ^a^, 0.006 ^b^
Tumor size	5.43 ± 4.00	5.83 ± 4.25	6.72 ± 5.01	not significant
UICC	1.80 ± 0.80 ^a^	1.90 ± 0.86 ^b^	2.23 ± 0.92 ^a,b^	0.002 ^a^, 0.036 ^b^
Steatosis grade %	20 ± 22	13 ± 17	11 ± 15	not significant
Inflammation grade	0.78 ± 0.61	0.62 ± 0.60 ^b^	0.90 ± 0.74 ^b^	0.026 ^b^
Fibrosis grade	4.07 ± 2.45	4.27 ± 2.34	4.05 ± 2.17	not significant
Age (years)	64.51 ± 11.61	64.08 ± 11.44	65.34 ± 10.98	not significant
Chemerin	1.59 ± 0.63 ^a,c^	2.14 ± 0.63 ^c^	2.38 ± 0.68 ^a^	<0.001 ^a^, 0.001 ^c^

## Data Availability

The data presented in this study are available on request from the corresponding author.

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
