# Peer review of "Chemerin and Chemokine-like Receptor 1 Expression Are Associated with Hepatocellular Carcinoma Progression in European Patients"

_biomedicines, 2023, doi:10.3390/biomedicines11030737_

Round 1

Reviewer 1 Report

The current study would like to provide evidence for ethnicity-related differences of hepatocellular carcinoma expressed chemerin and hepatocellular carcinoma severity.

Comments:

1. point 2.1. Patients: -Which is the age and the ethnicity of the patients? Please add this informations and than a discussion concerning this aspect - Explain please why did you prepared only 7 tissue micoarrays.   2. Results: -line 147 – on what basis do you make the following statement:”Tumors of females and males had comparable chemerin expression”. Please clarify. You wrote that the study is carried out on 383 patients, but the results are discussed only for a part of them. How do you explain this?   3.In conclusion, please state clearly what brings new the current study.

Author Response

We thank the reviewer for the helpful and fair comments. We want to inform the reviewer that sex-specific analysis has been included in the revised paper (please see tables 2, 3, 5, and 6) and the previous Figures 2 and 4 and all data related to the expression of chemerin and CMKLR1 in tumor and non-tumor tissues were deleted according to the comments of Reviewer 2. 

  1. point 2.1. Patients: -Which is the age and the ethnicity of the patients? Please add this information and than a discussion concerning this aspect - Explain please why did you prepared only 7 tissue micoarrays.  

Please see 2.1. of the revised manuscript:

“The patients are from the eastern part of Bavaria and about 2.5% of the inhabitants of Germany are from Asian countries. Mean age of the patients was 64.32 ± 11.48 years. Seven tissue microarrays (TMAs), in which up to 60 separate tissue cores per TMA were assembled, were prepared by using standard techniques already described”

Ethnicity was not recorded for each patient and this is now discussed in Limitations at the end of the discussion section

“The ethnicity of the patients with HCC was not recorded. The patients enrolled in the study are from the eastern part of Bavaria. About 2.5% of the German population are of Asian descent, and data of these patients could not be removed”

  1. Results: -line 147 – on what basis do you make the following statement:”Tumors of females and males had comparable chemerin expression”. Please clarify.

”Tumors of females and males had comparable chemerin expression” was corrected to “The chemerin staining scores of tumors of females and males did not differ (p = 0.156).” Statistical test used was Mann-Whitney U Test (to test for differences between two independent groups).

You wrote that the study is carried out on 383 patients, but the results are discussed only for a part of them.

This is unclear to us. In Table 1 and 4 data of all patients were used. Table 2 and 5 uses data of all females and table 3 and 6 data of all males.

  1. In conclusion, please state clearly what brings new the current study.

Conclusion was corrected as suggested by the Reviewer:

“High chemerin and CMKLR1 protein in European patients with HCC are related to adverse clinical parameters such as T-stage, vessel invasion, grading and UICC stage, indicating a tumor-promoting effect of this receptor and its ligand. In Asian patients, high tumor chemerin was protective. Preliminary analysis, moreover, indicates that the association of chemerin and CMKLR1 with HCC progression was less evident in female patients. Current findings suggest that ethnicity and sex have a role in HCC pathophysiology and have to be considered in the development of diagnostic and therapeutic approaches.  

Reviewer 2 Report

The aim of this original work was to „examine relationships between chemerin expression in HCC and the severity of HCC in a larger cohort of patients from Europe. CMKLR1 is a functional chemerin receptor, and its protein expression was analyzed in paralell.

The objectives and the introduction of the thesis are written clearly, I have no comments on this part of the thesis. Although the hypotheses of the thesis are expressed, I would recommend clearly writing the aim of the thesis also in the abstract.

Material and Methods: the study is retrospective, based on large tissue material from European HCC patients. The work is descriptive, based on a single research technique, that is, immunocytochemistry, which localises protein epitopes. A semi-quantitative scale was used for quantitative counts, which is highly subjective. These are the weaknesses of this work. Despite such a large number of tissues studied, clinical data are incomplete, there is no survival time of patients, no aetiology of HCC and no quantitative levels of these proteins e.g. in blood serum.

What was the control for the tissues with HCC was not described. From the description of fig. 2, it appears that the tissues accompanying the tumour?; there is no data regarding the "steatosis" evaluation scale. There is also no data on what the authors considered to be 'low', 'medium' and 'high' expression (table 1).

What p was considered statistically significant?

Results: lines 131-134 - I don't understand, where are these results? after all, one cannot draw such a conclusion on the basis of Figure 1.

In general, the presented figures are absolutely unacceptable, both in terms of description (histological image and ISH illegible, repeated data from material and methods, magnification inappropriate in fig. 1 and 2, 3 and 4 (all 200x?, unfortunately the "bar" value on the figures is not visible; there is no note that they are stained with haematoxylin, etc.). In the material available to the reviewer, the exact cellular localisation of the proteins sought is not visible. One sees mainly cell nuclei with a positive reaction, which, after all, was not subject to quantitative evaluation. Membranous expression is not visible in the magnifications shown. The shaded cells appear to have a cytoplasmic reaction, but this is uncertain. Comparisons of expression in patient 1 and 2 and 3 and 4 are unnecessary, it is unprofessional, such a 'by eye'.

The discussion is written according to the subject matter of the paper, but the results could have been related to practically two other papers [9-11] and an earlier paper by these authors [12]. I think the authors should have written clearly already in the introduction that the current study on chemerin/CMKLR1 is like a continuation of the earlier work and additionally investigates on a larger group of patients.

Minor comments:

1.      I disagree with some of the authors' terms, "the present experiments" (line 273)

2.      The description: „Immunohistochemistry of….” (lines 136, 169, etc.) are also incorrect, they should state: ‘Immunohistochemical expression…; or „Immunocytochemical localisation”…

3.      Please explain why there could be so much 'unspecific' staining, i.e. nuclear reaction of both proteins? Were different dilutions of antibodies tried?

To sum up:

1. the paper needs, above all, a revision of the photographic documentation of the results obtained. Please include fewer photographs (with possible removal of Figures 2 and 4), but with more convincing subcellular localisation (cytoplasm, cell membranes, etc.) and the cellular source of the expression (hepatocytes?, lymphocytes?, not „live”).

2. please also think about comparing the median values for the resulting semi-quantitative scale expression calculations.

3. please standardise the descriptions of all Figures and Tables.

4. please write that the work is a continuation of the authors' research, the aim of the work in the abstract, and what new information has been obtained as a result of the current research.

Author Response

We want to thank the reviewer for the very helpful comments on our manuscript.

The aim of this original work was to „examine relationships between chemerin expression in HCC and the severity of HCC in a larger cohort of patients from Europe. CMKLR1 is a functional chemerin receptor, and its protein expression was analyzed in parallel.”

The objectives and the introduction of the thesis are written clearly, I have no comments on this part of the thesis. Although the hypotheses of the thesis are expressed, I would recommend clearly writing the aim of the thesis also in the abstract.

This was corrected

“Studies from Asia found reduced expression of chemerin in HCC compared to para-tumor tissues while our previous analysis observed the opposite. Aim of this study was to correlate chemerin expression in HCC tissues with disease severity of European patients.”

Material and Methods: the study is retrospective, based on large tissue material from European HCC patients. The work is descriptive, based on a single research technique, that is, immunocytochemistry, which localises protein epitopes. A semi-quantitative scale was used for quantitative counts, which is highly subjective. These are the weaknesses of this work. Despite such a large number of tissues studied, clinical data are incomplete, there is no survival time of patients, no aetiology of HCC and no quantitative levels of these proteins e.g. in blood serum.

All of these limitations are now mentioned in Limitation of the study at the end of the discussion.

“Limitations of the current study are that survival was not documented and that disease etiology of most patients was unknown. Although positive associations of chemerin and CMKLR1 protein levels with markers of HCC severity suggest that higher levels of these proteins are linked to a worse prognosis, this has to be experimentally proven. Further limitations are the descriptive nature of this retrospective study, that only immunohistochemistry was used and that chemerin as well as CMKLR1 expression was semi-quantified by pathologist visual scoring of the staining intensity. The ethnicity of the patients with HCC was not recorded. The patients enrolled in the study are from the eastern part of Bavaria. About 2.5% of the German population are of Asian descent, and data of these patients could not be removed [18].”

Regarding serum we added the following paragraph in discussion

“Considering the close association of serum chemerin with hepatic dysfunction it is not unexpected that studies having measured circulating chemerin in HCC revealed discordant results. In European patients serum chemerin did not differ between HCC and controls [36]. A study from Japan could not find significant correlations of serum chemerin levels with HCC prognosis [35]. A study from China described a nearly 20-fold lower chemerin concentration in blood of HCC patients compared with healthy subjects [9]. It is well known that serum chemerin declines in patients with liver cirrhosis, and HCC occurs most often in patients with cirrhosis [33, 34, 37]. Future research has to compare serum chemerin levels between patients with and without HCC stratified for liver disease severity.“

We do not consider it a limitation that serum chemerin was not analyzed because this was not the aim of our study.

What was the control for the tissues with HCC was not described. From the description of fig. 2, it appears that the tissues accompanying the tumour?; there is no data regarding the "steatosis" evaluation scale. There is also no data on what the authors considered to be 'low', 'medium' and 'high' expression (table 1).

Please see 2.2. “The control group consisted of non-neoplastic liver tissue from patients suffering from HCC.” This is now also mentioned in legend of figure 1 and 2.

Grading of steatosis was now better explained: “For grade of tumoral steatosis, the percentage of tumoral fat vacuoles in relation to total tumor volume was stated. Steatosis grade ranged from 0% to 80%.”

Please see 2.2. for definition of scores

“Low staining intensity was defined as no or barely visible membranous and/or cytoplasmic staining, for high staining intensity either heterogeneous or patchy membranous and/or cytoplasmic staining comparable to that of bile ducts in normal liver tissue was considered and medium staining intensity was defined as any staining intensity between the other two groups.

A three-tiered scoring system was used for chemerin, with scores 1 denoting low cytoplasmic and/or membranous staining, 2 denoting moderate staining, and 3 denoting high staining. A three-tiered score system was also used to define the level of CMKLR1 protein expression, with scores 1 and 2 designating low, moderate, and 3 designating high cytoplasmic and/or membranous staining, respectively.“

What p was considered statistically significant?

Please apologize this mistake. Please see 2.4. “A p-value < 0.05 was considered significant.”

Results: lines 131-134 - I don't understand, where are these results? after all, one cannot draw such a conclusion on the basis of Figure 1.

We fully agree with your opinion. All the data comparing chemerin and CMKLR1 in HCC and adjacent tissue were deleted.

In general, the presented figures are absolutely unacceptable, both in terms of description (histological image and ISH illegible, repeated data from material and methods, magnification inappropriate in fig. 1 and 2, 3 and 4 (all 200x?, unfortunately the "bar" value on the figures is not visible; there is no note that they are stained with haematoxylin, etc.). In the material available to the reviewer, the exact cellular localisation of the proteins sought is not visible. One sees mainly cell nuclei with a positive reaction, which, after all, was not subject to quantitative evaluation. Membranous expression is not visible in the magnifications shown. The shaded cells appear to have a cytoplasmic reaction, but this is uncertain. Comparisons of expression in patient 1 and 2 and 3 and 4 are unnecessary, it is unprofessional, such a 'by eye'.

We have no included figures with a higher magnification. The original scale bar is indeed hardly visible (at the lower left edge) and thus we included a new bar (at the upper left edge) corresponding in size to the original one.

Data related to the comparison of HCC and HCC adjacent tissues were deleted.  

In 2.2. it is indicated “After these incubations, the slides were counterstained with hematoxylin.” This information is now also given in the figure legends.

The discussion is written according to the subject matter of the paper, but the results could have been related to practically two other papers [9-11] and an earlier paper by these authors [12]. I think the authors should have written clearly already in the introduction that the current study on chemerin/CMKLR1 is like a continuation of the earlier work and additionally investigates on a larger group of patients.

All the data comparing chemerin and CMKLR1 in HCC and adjacent tissues were deleted. Because this was the main topic of our previous study we do not refer to this investigation in great detail in the Introduction.

To address your comment we added in the Introduction:

“In contradiction with these findings from China our previous analysis of chemerin protein levels by immunoblot in HCC tissues obtained from European patients detected higher protein in HCC than the para-tumor tissues”

“Upregulation of chemerin in HCC tissues of Europeans suggests a tumor-promoting function, but associations of hepatocyte expressed chemerin and HCC progression in this population have not been studied so far.  “

Minor comments:

  1. I disagree with some of the authors' terms, "the present experiments" (line 273)

This was corrected.

  1. The description: „Immunohistochemistry of….” (lines 136, 169, etc.) are also incorrect, they should state: ‘Immunohistochemical expression…; or „Immunocytochemical localisation”…

This was corrected.

  1. Please explain why there could be so much 'unspecific' staining, i.e. nuclear reaction of both proteins? Were different dilutions of antibodies tried?

Please see 2.2.

“The IHC-plus™ antibodies are tested in immunohistochemistry against human formalin-fixed paraffin-embedded tissues and have an excellent specificity and sensitivity for detecting the target protein [21]. The staining protocol was established as is done for all antibodies in the Institute for Pathology by trying out different dilutions of the antibody and different pre-treatments on a control TMA with the most important normal human tissues until the optimal staining protocol with a balance between staining intensity of the desired protein and minimal background staining is found“

Considering unspecific staining this was now also discussed:

“Membrane-cytoplasmic staining of CMKLR1 was described in mouse liver [37]. The CMKLR1 antibody distributed by a different company showed cytoplasmic and nuclear staining in kidney carcinoma and stomach carcinoma [38]. Nuclear staining was also observed in the current experiments but was regarded as unspecific and not quantified. It is possible that CMKLR1 localizes to the nucleus in different tumor cells but further experiments have to prove this assumption. Nuclear staining of chemerin is considered unspecific, and there are no studies having shown that chemerin translocates to the nucleus as far as we know. ”

To sum up:

  1. the paper needs, above all, a revision of the photographic documentation of the results obtained. Please include fewer photographs (with possible removal of Figures 2 and 4), but with more convincing subcellular localisation (cytoplasm, cell membranes, etc.) and the cellular source of the expression (hepatocytes?, lymphocytes?, not „live”).

The paper was revised according to the comments of the reviewer. It is now clearly indicated that only staining of hepatocytes was scored.

Figure 2 and 4 and all data related to the expression of chemerin and CMKLR1 in tumor and non-tumor tissue were deleted.

  1. please also think about comparing the median values for the resulting semi-quantitative scale expression calculations.

This comment is unclear to us. We used a non-parametric test for comparison of more than two unrelated groups. This is the correct statistical test which The determines whether the medians of two or more groups are different. We listed the mean value ± standard deviation because the median values are relatively similar when calculating scores ranging from 1 to 3. This is clearly described in 2.4.

  1. please standardise the descriptions of all Figures and Tables.
  2.  

Description of figure 1 and 2 is now standardized

  1. please write that the work is a continuation of the authors' research, the aim of the work in the abstract, and what new information has been obtained as a result of the current research.

Figure 2 and 4 and all data related to the expression of chemerin and CMKLR1 in tumor and non-tumor tissue were deleted. Therefore, this work is – in our opinion - not really a continuation of our previous study.

Round 2

Reviewer 2 Report

I have read the submitted manuscript after review and I believe that in its current form the work can be accepted for publication. It seems that all my comments have been taken into account and the work is now presented in a more professional manner.

Regarding the comment about presenting results in Tables in the form of mean value and SD. I agree that this is a valid statistical test to determine whether the medians of two or more groups are different. Nevertheless, it would be more correct to present the median values, min. and max. values, values of mode along with the mean values ± standard deviation in the Tables. This is the case for statistical characteristics evaluated on semi-quantitative scales, as in the present work.

If there were any minor proofreading errors (extra spaces, missing periods, etc.) that I did not notice, I hope they will "disappear" after the final brush-up.